# Design and Experimental Characterization of Developed Human Knee Joint Exoskeleton Prototypes †

## Michał Olinski

Department of Fundamentals of Machine Design and Mechatronic Systems K61W10D07, Wrocław University of Science and Technology, Łukasiewicza St. 7/9, 50-371 Wroclaw, Poland; michal.olinski@pwr.edu.pl;
Tel.: +48-713202710

† This article is an expanded version of a paper entitled An experimental characterization of developed knee joint mechanism prototypes, which was presented at MEDER 2024, Timisoara, Romania, 27–29 June 2024.

**Abstract:** This paper focuses on the experimental testing and characterisation of two designed and constructed prototypes of a human knee joint mechanism. The aim of the mechanical systems, presented as kinematic diagrams and 3D CAD drawings, is to reproduce the knee joint's complex movement, in particular the flexion/extension in the sagittal plane, within a given range and constraints, while taking into account the trajectory of the joint's instantaneous centre of rotation. The first prototype can simulate different movements by modifying its dimensions in real time using a linearly adjustable crossed four-bar mechanism. The second prototype has interchangeable cooperating components, with cam profiles that can be adapted to specific requirements. Both devices are built from 3D-printed parts and their characteristics are determined experimentally. Although many types of tests have been carried out, this research mainly aims to conduct experiments with volunteers. To this end, the IMU sensors measure the mechanisms' movements, but the main source of the data is video analysis of the colour markers. For the purposes of postprocessing, the results in the form of numerical values and figures were computed by Matlab 2019b. To illustrate the prototypes' capabilities, the results are shown as motion trajectories of selected tibia/femur points and the calculated knee joint's flexion/extension angle.

**Keywords:** biomechanics; experimental mechanics; rehabilitation; video analysis; centroid; prosthesis; orthosis; knee articulation; polycentric hinge; IMU

## 1. Introduction

The human knee comprises two main parts: the patellofemoral joint and the tibiofemoral joint. The first of these plays a vital role in knee extension by enhancing the leverage of the quadriceps muscles, but it is the tibiofemoral joint that is the primary weight-bearing joint crucial for knee stability and movement function. Therefore, knee joint movement is controlled and limited by the meniscus between the femur and tibia, the shapes of their surfaces that form the human knee joint, and a collection of ligaments (mainly the cruciate ligaments). This movement is complex because, in addition to the main rotation of the shin relative to the thigh, there is also an additional translational movement caused by the slide. Therefore, although commonly simplified to a simple 1DOF hinge, the knee has two degrees of freedom (2DOFs) even when only flexion/extension in the sagittal plane is considered (Figure 1a). For these reasons, the instantaneous

centre of rotation (ICR) of the joint is often used to characterise knee joint movement. It changes position as the angle of flexion changes [1,2]. The characterisation also includes trajectories of selected tibia/femur points [3].

It is common practice to determine the characteristics of knee joint movement and its various developed mechanisms through experiments, using a variety of developed tools and methods. These include cameras with attached reflective markers [4,5] or colour markers [6,7], which are used for real-time monitoring [8] and movement characterisation [9]. Similar tasks can be performed using Kinect to recognise and track specific colours in images [10]. Tools such as electrogoniometers [11] and inertial measurement units (IMUs) [12–14] are also used as alternatives.

In some reasonable cases, certain devices that replicate knee joint movement reduce it to a hinge. Examples of such devices are rehabilitation manipulators [15,16] and numerous continuous passive motion machines (CPMs) [17,18]. However, it is necessary to consider this complex movement in certain cases, such as prostheses and most joint rehabilitation devices. These devices aim to restore the normal range of motion (ROM), muscle and nervous system function, as well as to promote tissue, bone and cartilage regeneration by subjecting them to naturally occurring loads. Overlooking this complex movement can result in ineffective treatment or unexpected detrimental effects on the patient's health.

Consequently, many knee mechanisms, including some spatial prostheses [19], have been developed to reproduce the complex action. However, in most of them the movement is limited to the sagittal plane and makes use of gears [20,21] and the four-bar mechanism [22], among others. Some devices use the non-crossed four-bar mechanism [23,24]. In contrast, other devices use a crossed four4-bar mechanism, such as in [25–29] and [30], where the exoskeleton orthosis is exposed (Figure 1b) or in [31] proposing a more complex solution (a single-DoF combined mechanism).

In order to similarly replicate natural movements, other solutions often rely on the so-called "bionic knee", which mimics the structure of the human joint. While some of these solutions make use of bone shapes, like endoprosthesis (DAEQOUS-G1) [32], others use cams and grooves [2], and in some cases the meniscus and patella are also considered [33]. However, as joint movements are individual, the geometry of these devices must be adapted to each user.

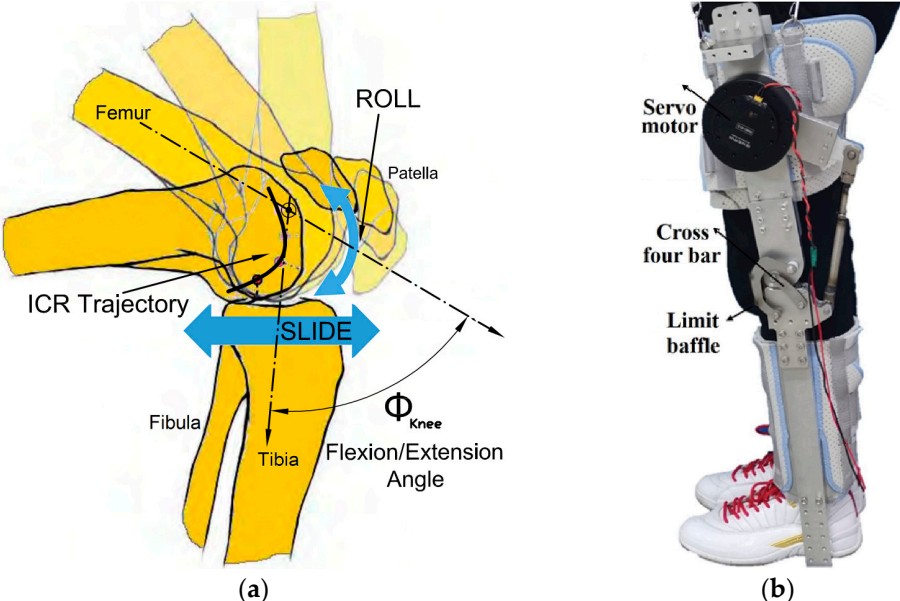

(**a**)          (**b**)

**Figure 1.** Knee joint: (**a**) complex movement with an ICR trajectory and the joint angle simplified to 1DOF in the sagittal plane [12]; (**b**) the crossed 4-bar mechanism in the exoskeleton [30].

Due to the aforementioned factors, many researchers tend to focus on designing and analysing human knee joint mechanisms that can replicate its complex movements. The main contribution and objective of this article is then to investigate two designed novel knee joint mechanisms, and their feasibility in terms of reproducing this complex motion. Following up the research presented at the MEDER24 conference (Timisoara, Romania) [34], this study focuses on presenting the results of an experimental evaluation of the preliminary exoskeleton prototypes: a dampers prototype with an adjustable crossed four-bar mechanism and a cam prototype with interchangeable customizable components. CAD software (Autodesk Inventor 2018) was used to create three-dimensional models of the mechanisms including their defined kinematics, geometry and cam profiles. The subsequent paragraphs encompass the design of the mechanisms, the constructed prototypes and their characteristics using data from physical trials of the devices. Particular emphasis is placed on the video analysis (the tracking of colour markers) of trials conducted at a stationary frame and with volunteers. To characterise the prototypes, the trajectories of specific femur/bone points and knee joint flexion/extension angles were developed and presented.

## 2. Materials and Methods

### 2.1. Knee Joint Mechanism Prototypes

Two preliminary exoskeleton prototypes of human knee joint mechanisms were designed and developed. Both aim to accurately reproduce the complex movement of the knee joint in the sagittal plane within the assumed range and limitations, by using different techniques. Therefore, the focus of this study is placed on the tibiofemoral joint, while the other knee elements, like the patellofemoral joint, are considered for their function, but to a limited extent. The prototypes were designed with a specific exemplary ICR trajectory based on the literature [25].

The first prototype (the dampers prototype) draws on a linearly adjustable crossed 4-bar mechanism (Figure 2a). Its size can be changed in real time to mimic different ICR trajectories, due to the additional prismatic joints (2 additional DOFs). The cam mechanism is used in the second prototype, named the cam prototype (Figure 2b). Fixed and moving centroids of the ICR of the knee joint serve as a model for the cooperating cam profiles [7].

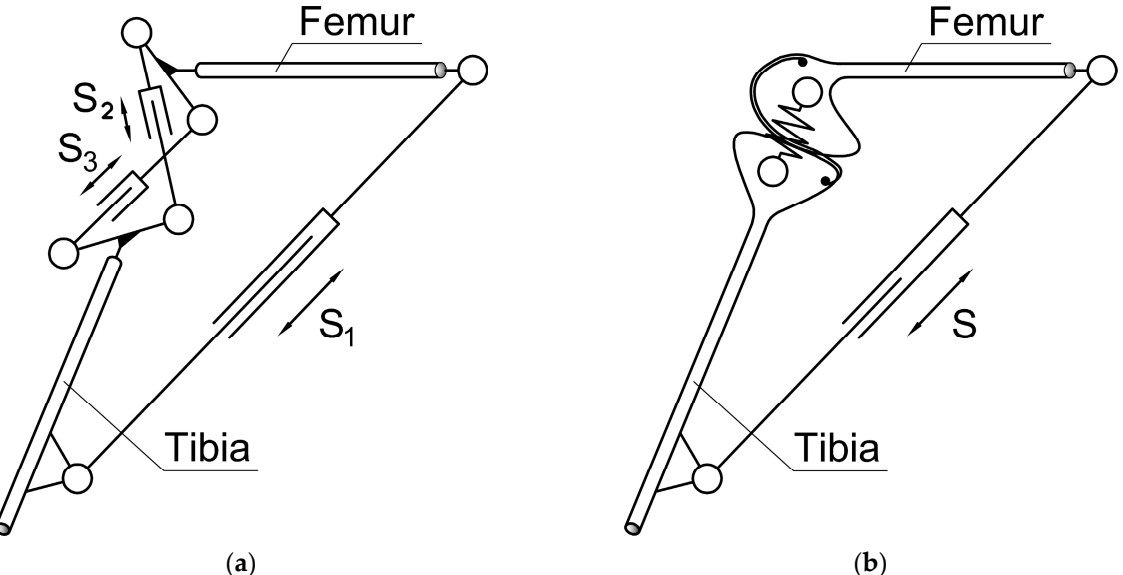

(**a**)  (**b**)

**Figure 2.** Two proposals for knee joint mechanisms shown in kinematic diagrams: (**a**) dampers prototype based on linearly adjustable 4-bar mechanism; (**b**) cam prototype based on cam mechanism.

Detailed 3D models of the prototypes were created using Autodesk Inventor 2018 software, taking into account their close interaction with the human knee (Figure 3a—the cam prototype model). Both prototypes were built with 3D-printed parts and screws. They were equipped with IMU sensors and coloured markers to track the movement of the different parts on video captured during the experiments. In addition, the cam prototype has ties (inserted into the guide grooves—Figure 3a) to ensure smooth rolling between cams without slippage, and a uniquely shaped metal element that acts as a spring. The cam elements are interchangeable and, because they are 3D-printed, they can be easily adapted to specific requirements and achieve a range of necessary trajectories. Table 1 presents some further design and technical details, i.e., the overall dimensions at full extension and the determined initial dimensions of the 4-bar mechanism. However, since the built devices are prototypes, some features like the critical tolerances of particular dimensions can be determined only after the final forms of the devices have been developed and the manufacturing drawings of individual parts have been prepared. At this stage of the research, it was possible to identify that the crucial dimensions would be the points of dampers connections and, for the second prototype, the exact shapes of the cooperating cam profiles.

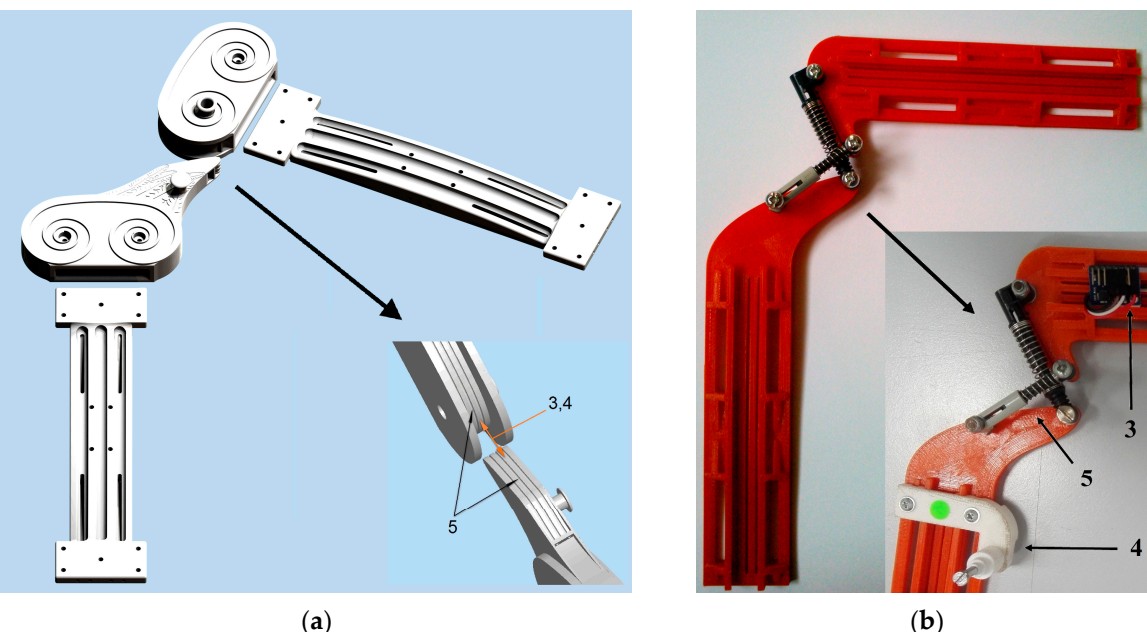

(**a**)  (**b**)

**Figure 3.** Developed knee joint mechanisms: (**a**) the three-dimensional cam prototype with an additional magnified view that clarifies the principle of operation by showing the cooperating profiles of the cams (3, 4) and the guide grooves (5); (**b**) the functional dampers prototype built using the dampers (5), with a visible (in the magnified view) IMU sensor (3) and the white element (4) used to fix the linear motor.

**Table 1.** The prototypes' design details including 3D printing parameters.

| No. | Prototype | Main Dimensions (Length/Width) | Critical Dimensions (Figures 2a and 3a) | Printing Material | Printing Layer Thickness |
|---|---|---|---|---|---|
| 1. | Dampers prototype | Overall 360 mm/80 mm Tibia 195 mm/40 mm | $l_F$ = 38.8 mm (element at femur), $S_2$ = 48.6 mm, $S_3$ = 43 mm, $l_T$ = 37.3 mm (element at tibia) | PLA | 0.3 mm |
| 2. | Cam prototype | Overall 405 mm/95 mm Tibia 225 mm/40 mm | exact dimensions of cooperating cam profiles matching the desired ICR centroids [25] | PET-G | 0.2 mm |

Figure 3b shows a linearly adjustable knee joint mechanism as a complete dampers prototype. Crossed bars with prismatic joints are constructed using common dampers.

## 2.2. Experimental Layout and Performed Experiments

A number of tests were carried out using both constructed prototypes. Table 2 provides further details and information on the variables that were tested. The tests used both powered prototypes in a stationary frame (modes 1, 2) and prototypes worn by human volunteers (modes 3, 4). Among the data collected were video recordings of the markers' movements and the triaxial orientation of the IMU sensors ($\Theta_{x,y,z}$), angular velocity ($\omega_{x,y,z}$), linear acceleration ($a_{x,y,z}$) and, for the stationary option, $P_c$ energy consumption. The useful output parameters include $x(t)$, $y(t)$ and $y(x)$—the markers' displacements and trajectories; $\Theta_{knee\_M}$, $\Theta_{knee\_I}$—the knee joint flexion/extension angle determined by markers/IMUs; and drive force F (the stationary option).

**Table 2.** The experimental modes of the trials conducted and the collected variables.

| No. | Experimental Modes | Saved Variables | | Output Parameters | |
|---|---|---|---|---|---|
| 1. | Frame, dampers prototype | Video of the markers movement | F | From markers: | $P_C$ |
| 2. | Frame, cam prototype | From IMU | — | displacements x, y and trajectories y(x) | — |
| 3. | Human, dampers prototype | $\Theta_{x,y,z}$ $\omega_{x,y,z}$ $a_{x,y,z}$ | | $\Theta_{knee\_M}$, $\Theta_{knee\_I}$ | |
| 4. | Human, cam prototype | | | | |

A laboratory experimental rig was used for the experiments, the layout of which is shown in Figure 4; the components are numbered as: (1) white screen; (2) two versions of the experiment, (2a) knee prototype mounted on a stationary rig, (2b) knee prototype attached to a volunteer; (3) IMU sensors attached to the prototype; (4) linear motor to force flexion/extension of the prototype; (5) video recording device; (6) energy consumption sensor; (7) colour markers to track movement after video processing.

As shown in Figure 5a,b, the knee prototypes for modes 1 and 2 were attached to a stationary frame. The coordinate systems used for the calculations are shown in the second view. A linear motor was used to force 10 repetitions of a flexion/extension movement, with 0.5 s stops at extreme positions.

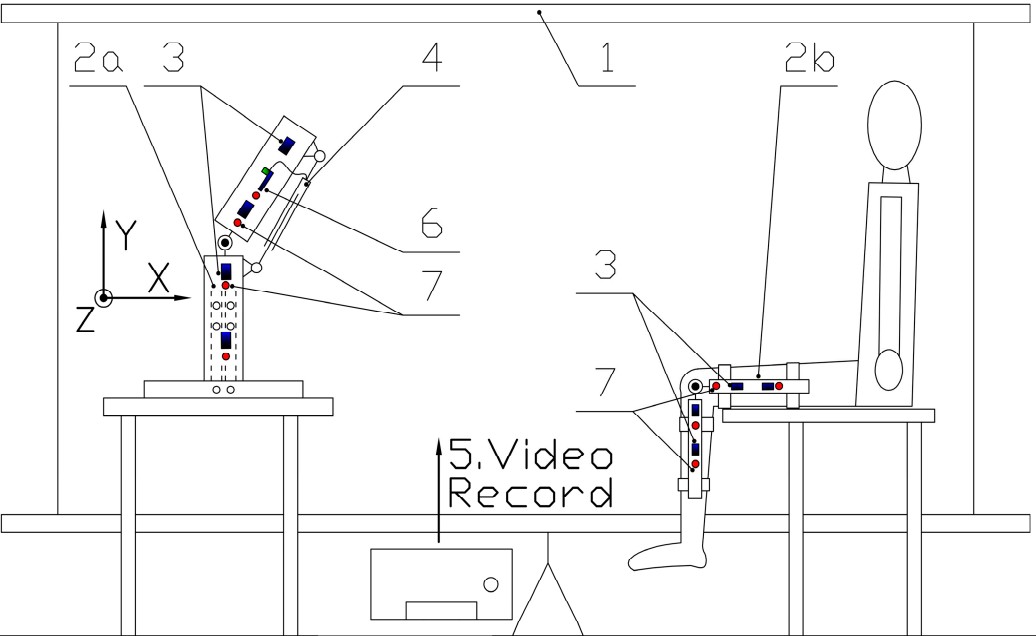

**Figure 4.** A diagram showing the experimental rig in the laboratory, with numbers denoting the different components of the prototypes and the bench. Two test versions are shown: one with a stationary rig (**left**) and one with human volunteers (**right**).

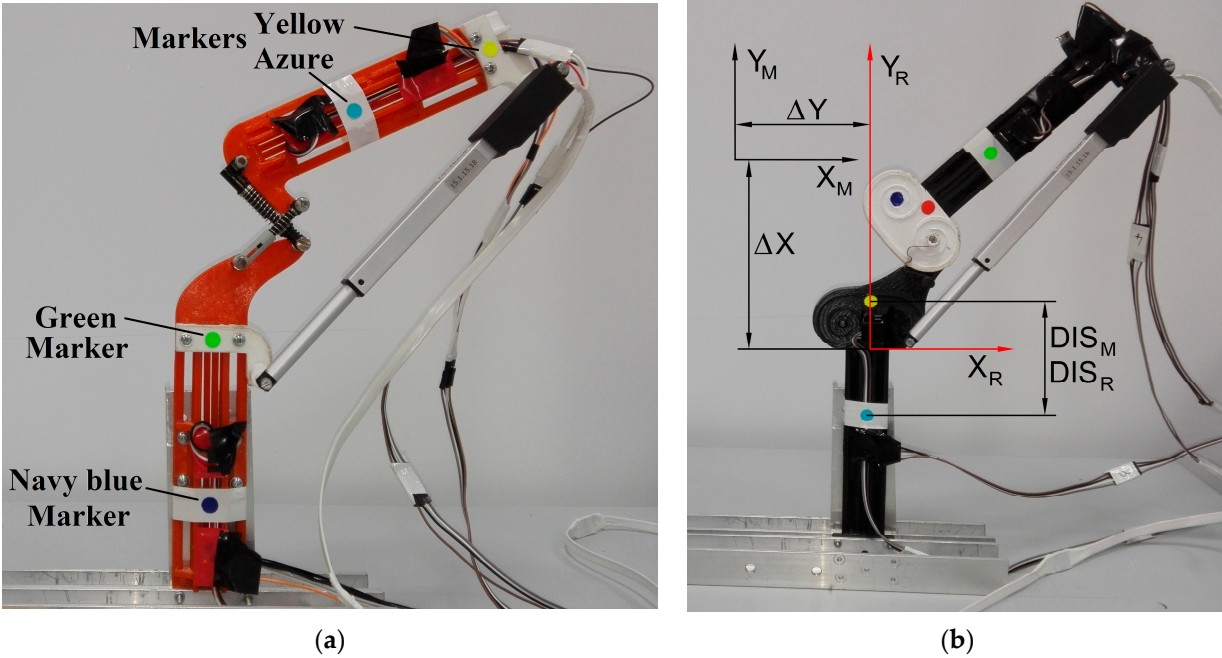

(**a**)                                                              (**b**)

**Figure 5.** Physical prototypes of knee joint mechanisms installed on a stationary test rig frame with an actuator, markers and sensors: (**a**) the dampers prototype with a linearly adjustable crossed 4-bar mechanism; (**b**) the cam prototype with plotted coordinate systems used for movement identification and calculations [7].

The two volunteers were male and female, both in the age range of 25–30 years old. They took part in the tests for both prototypes in modes 3 and 4, as shown in Figure 6a,b. At least three trials were conducted for each prototype and volunteer, involving ten repetitions of the flexion/extension movement. The prototype was attached with Velcro straps to the volunteer's left lower limb.

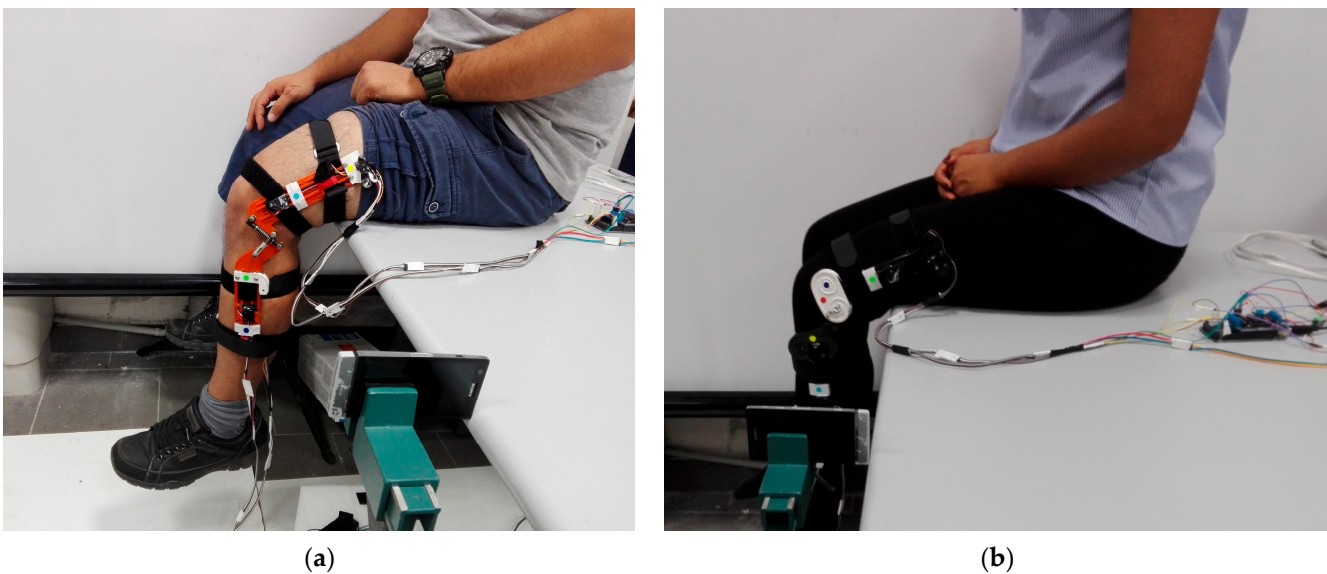

(**a**)                                                              (**b**)

**Figure 6.** Volunteers wearing knee joint prototypes on the experimental rig: (**a**) the dampers prototype on the first volunteer; (**b**) the cam prototype on the second volunteer.

All of the performed trials followed the experimental procedure/protocol presented as a block diagram in Figure 7. This began with checking the IMU sensors and measure-

ment system, and the experimental layout was set up according to the current experimental mode, placing each component in the appropriate location. As part of the preparations, the volunteers were requested to sit in the position shown in Figure 6, while maintaining the distance of the knee joint from the edge of the seat to prevent the restriction of the range of motion. The position and orientation of the prototype was then determined to match the volunteer's thigh and tibia. Then, the main part of the trial was the performance of movements following the adopted mode. The equipment was calibrated at the beginning of each session. Particularly for the IMU sensors, to correct the zero error of the sensors' axes of initial orientation, an online calibration was also performed. Specifically, the IMU results from 10 s of a motionless measurement position were averaged. Video recordings were taken, and sensor measurements were recorded during the execution of the movements; the trials ended with arranging and interpreting the data.

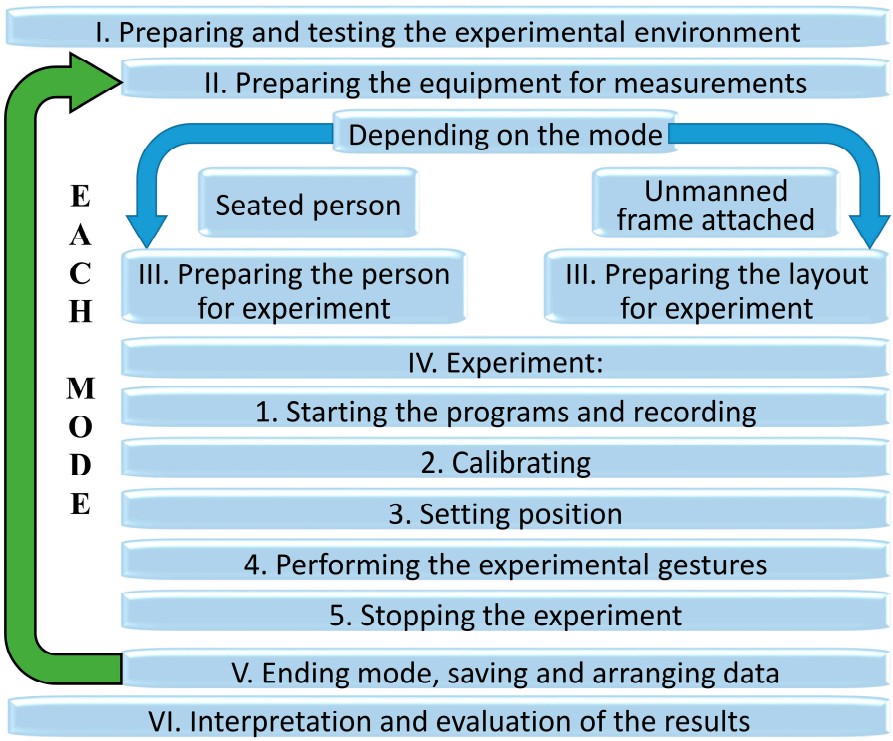

**Figure 7.** A block diagram presenting the experimental protocol.

*2.3. Measurement System and Postprocessing*

By creating a data flow diagram, as shown in Figure 8, the main operational algorithm of the measurement system was established. The two main sources of movement data are the camera recordings and the IMU sensors.

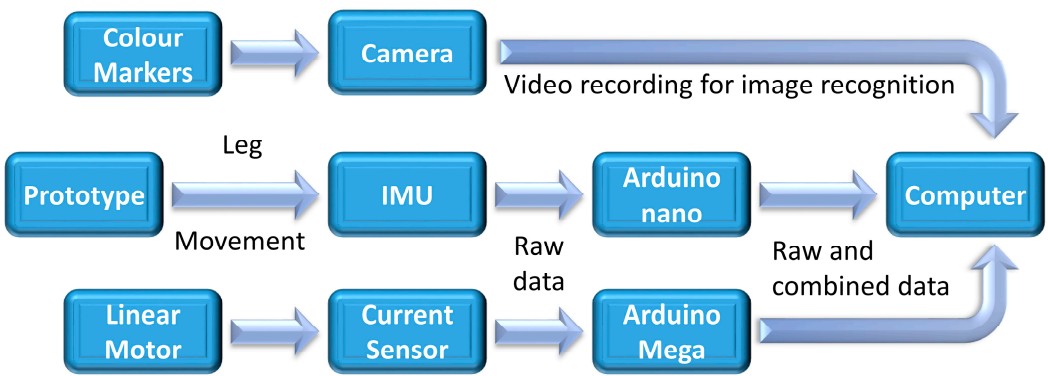

**Figure 8.** A block diagram of the data flow for the performed experiments.

As mentioned earlier, in order to visually analyse the movement of the mechanism and assess its performance, coloured markers were placed on the prototypes: two on the femur and two on the tibia. To track the locations of the markers, the collected movement recordings from the trials were processed using Matlab's Computer Vision System Toolbox.

Individual images (frames) were extracted from the recordings taken during the experiment at a rate of 30 frames per second. Each was examined to identify the colour markers defined in RGB as averaged over a marked circular area (Figure 9a). The settings of the analysis program, in particular the range of colour shades, had to be changed for each experiment and marker colour (Figure 9b, misrecognition) to enable the accurate detection of marker displacement (Figure 9c). A similar procedure was applied for each experimental mode, enabling the tracking of the x and y positions of the markers, relative to the coordinate system defined in MATLAB (Figure 5b—black $X_M$, $Y_M$). The use of a scaling factor also allowed for the results to be aligned with the value of the actual movement. This was achieved by comparing the physical distance ($DIS_R$) between two colour markers on the prototype when the tibia is immobile (Figure 5b), and the measured distance between these markers in MATLAB ($DIS_M$). To obtain locations according to the origin of the coordinate system on the physical prototype (Figure 5b—red $X_R$ and $Y_R$), the resulting positions were further recalculated (using $\Delta X$ and $\Delta Y$ according to Figure 5b).

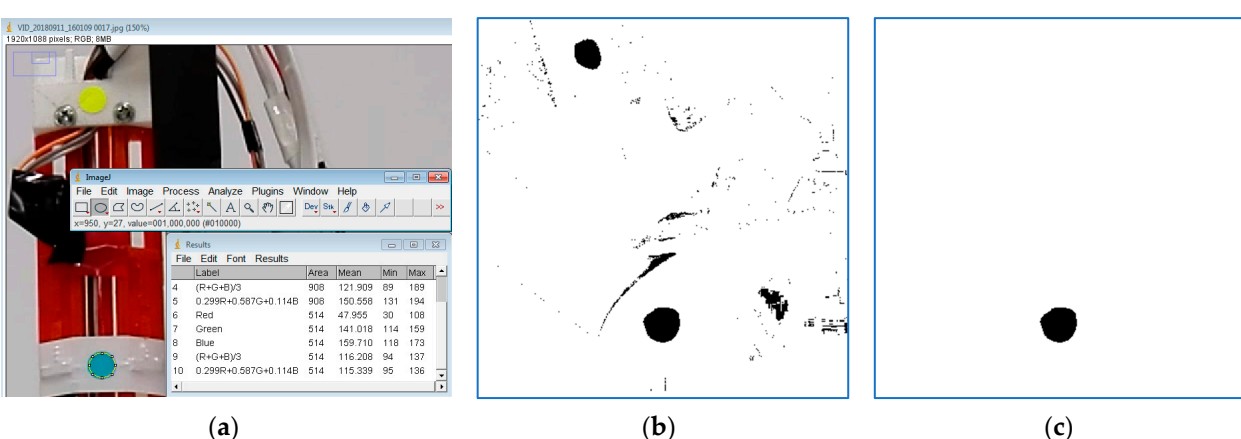

(**a**)  (**b**)  (**c**)

**Figure 9.** Image analysis—marker recognition: (**a**) the determination of the average RGB values of the colour of the marked (circle) area under investigation; (**b**) the recognition of multiple areas if the range of possible shades was too wide; (**c**) a correctly recognised marker with a defined range of colour shades (no other recognised elements).

Additionally, IMU sensors (GY-85 Arduino board) were mounted on both prototypes: two on the femur and two on the tibia. Each sensor includes a HMC5883L magnetometer, an MPU6050 with an ADXL345 accelerometer (with a range of ±8 g) and an ITG3200 gyroscope (all from Wobit company, Gdańsk, Poland). The frequency of measurement was set to 50 Hz and the foundation for accurate sensor calibration was [35]. The IMUs recorded 9 variables as raw data in their local 3D coordinate system (Figure 10a), including linear acceleration, angular velocity and the magnetic field vector. Finally, Figure 10b shows the combined data: Euler angles ($\Psi$, $\theta$, $\Phi$—yaw, pitch, roll) characterise the sensor orientation. The sensor fusion algorithm (the Direct Cosine Matrix method) [35] is applied to raw measured data to obtain these angles.

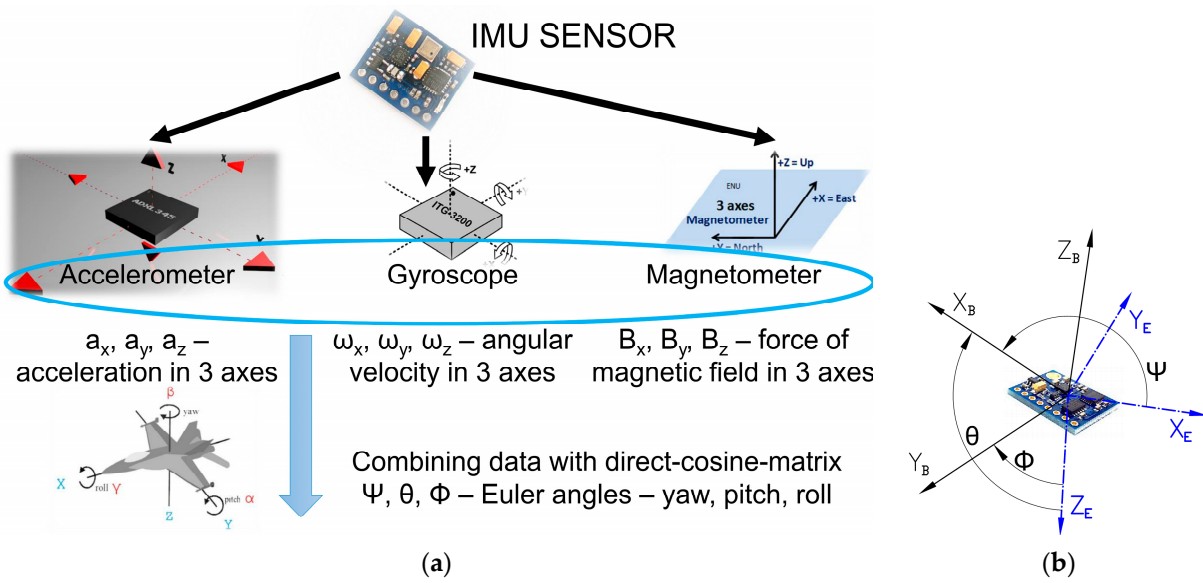

**Figure 10.** The used IMU sensor: (**a**) a schematic representation of the operating principle, the raw data collected and the combined data in the form of Euler angles; (**b**) the electronic element with a scheme of the measured Euler angles [12].

## 3. Results

As reported earlier, a series of experiments were conducted using powered prototypes at a stationary rig (modes 1 and 2) and with volunteers wearing both prototypes (modes 3 and 4). These were performed according to the developed experimental protocol. For the stationary rig, part of the obtained results is presented mostly for reference, since this article concentrates predominantly on the experiments with volunteers.

### 3.1. Results for Prototypes at a Stationary Rig

The data collected from the video analysis of the two colour markers, azure and yellow, placed on the femur of the dampers prototype (the tibia is immobile) are shown in Figures 11 and 12. After the postprocessing described earlier, the movements of the markers were obtained and are presented, among others, as vertical displacements y(t) against time (Figure 11). The minimum/maximum values of the x and y positions of the markers obtained in successive repetitions are similar, with variation between repetitions not exceeding ±0.3 mm.

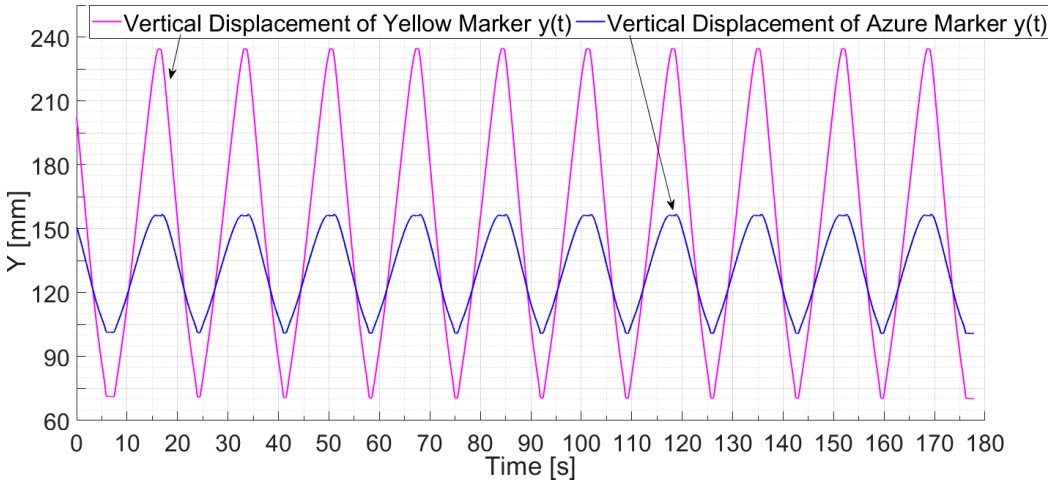

**Figure 11.** The vertical displacement y(t) of the markers placed on the thigh—dampers prototype [27].

In addition, the diagrams of movement trajectories y(x) for the azure and yellow markers are presented for the dampers prototype (Figure 12). Similarly, the results of movements were obtained for the cam prototype and are presented as the trajectories of the red and green markers of the femur (Figure 13). For subsequent repetitions, the type of movement for both prototypes was maintained, which yields almost identical trajectory curves.

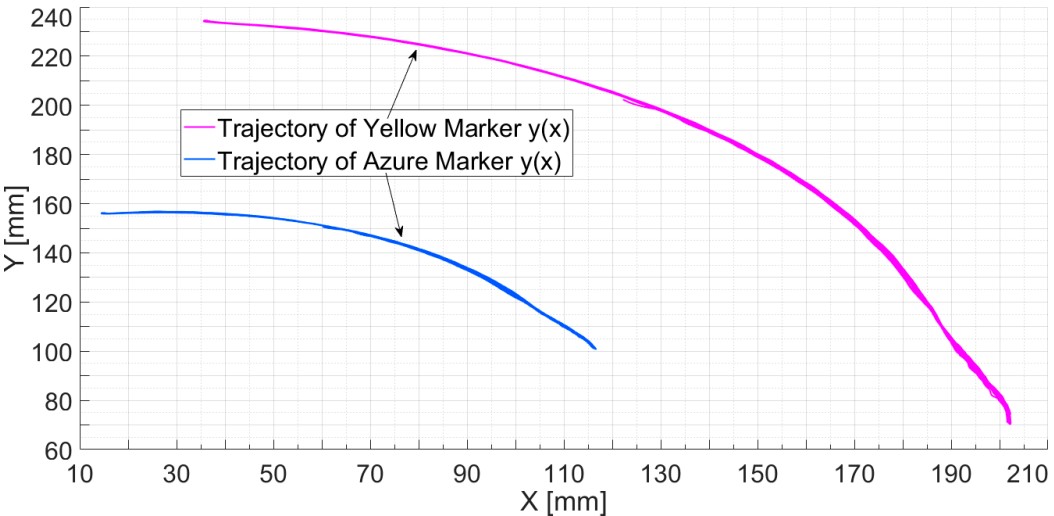

**Figure 12.** The trajectory of movement y(x) of the azure and yellow markers—dampers prototype [27].

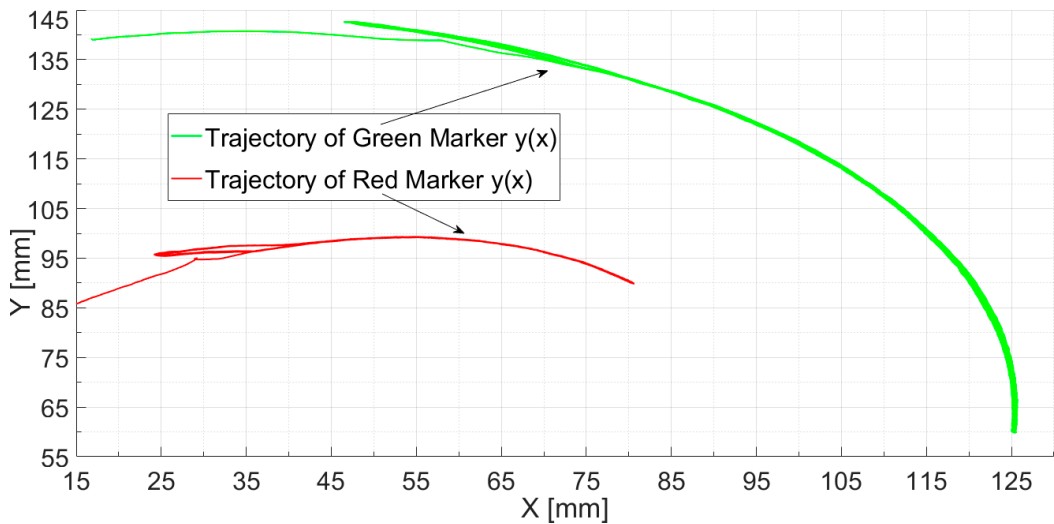

**Figure 13.** The trajectory of movement y(x) of the red and green markers—cam prototype [7].

In addition, the angular orientation of the femur, which in this case can be equated with the clinically relevant parameter of the flexion/extension angle of the knee joint, was determined by combining the displacement data of the two mobile markers. Thus, reproducible and satisfactory movement results were obtained. For the dampers prototype (Figure 14), the maximum flexion and extension angles achieved by the knee joint in successive repetitions of the movement were 108.5° to 108.4° and 14.03° to 13.98°, respectively, resulting in variation of ±0.05° and ±0.025°. The angles obtained for successive cam prototype movements (Figure 15) range from approximately 24° to 122°.

Subsequent repetitions of the experimental flexion/extension movement showed an overall high repeatability of the prototype movement, both for the angular ROM measurement (Figures 14 and 15) and the displacements of the markers (Figures 11–13).

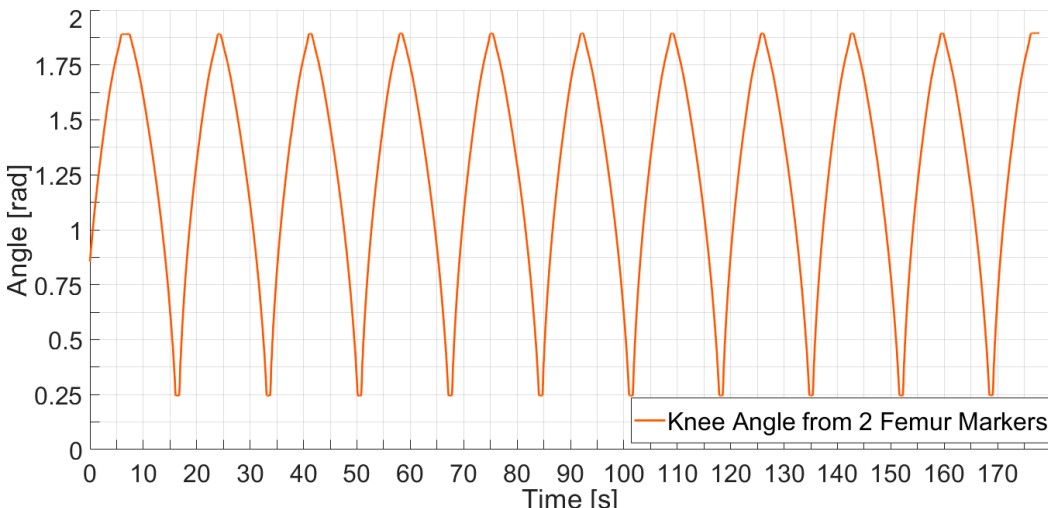

**Figure 14.** The flexion/extension angle of the knee joint based on the movement of two thigh markers–dampers prototype [27].

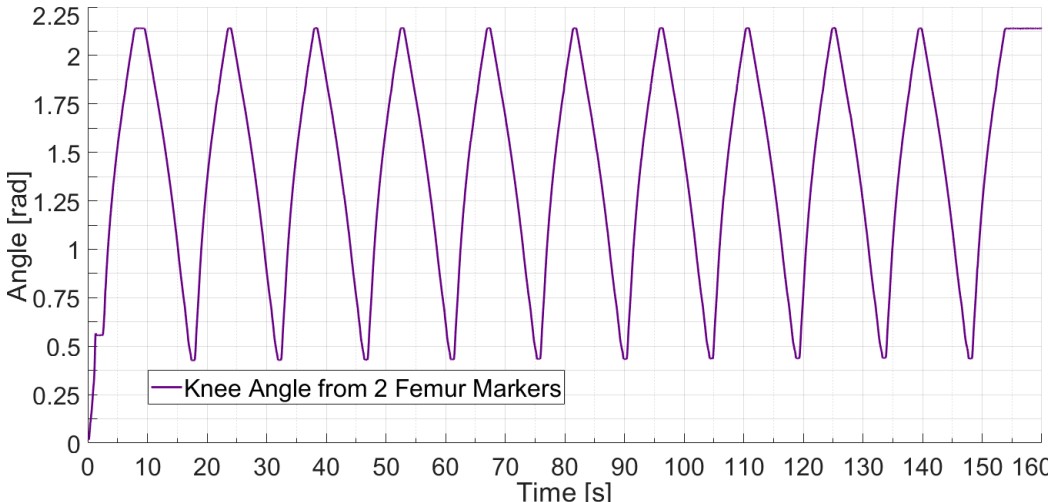

**Figure 15.** The angle of the knee joint based on the movement of the femur markers—cam prototype [7].

### 3.2. Results for Prototypes Worn by Volunteers

Due to minor problems with the correct identification of the selected colour shades of the markers, such as navy blue (close to full knee extension), a small proportion of the data has been approximated for the volunteer trials. The x and y movements (Figure 16) and consequently the y(x) trajectories (Figures 17 and 18) of the tibia points were then obtained, constituting both prototypes' characteristics. For both prototypes, the obtained minimum and maximum values of the markers' positions (excluding a few major errors in the calculations at the beginning and end of the movement) are approximately (in mm): green(x) −75.00/24.08, (y) 22.72/144.85; yellow(x) −167.36/73.30, (y) −67.83/111.54 (dampers prototype); yellow(x) −12.21/16.60, (y) 55.92/95.82; azure(x) −81.67/48.93, (y) −26.09/53.20 (cam prototype).

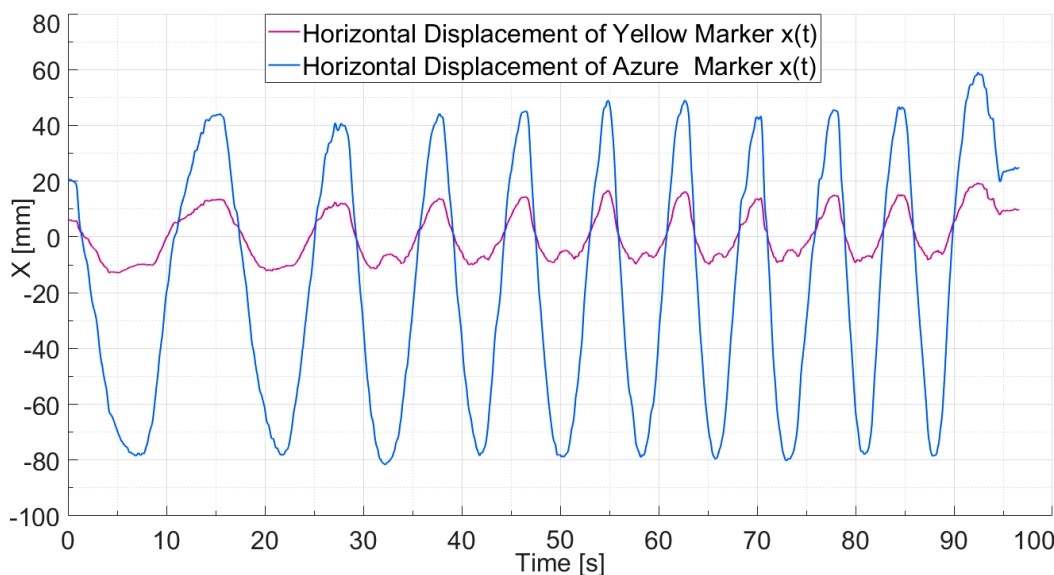

**Figure 16.** X(t) as the horizontal movement of the azure and yellow markers on the tibia—cam prototype.

For the volunteers' trials, the flexion/extension angle of the knee joint in the sagittal plane is presented as the final result (Figure 19). In this case, it is calculated as the difference in angle between the tibia and femur. Despite the seated position, the femur's movement was significant and could not be omitted. These results are compared with the angle obtained from the IMU sensors (Figure 19a,b). The minimum and maximum values obtained for the knee joint are approximately 2.06° ÷ 95.32° (0.04 rad ÷ 1.66 rad, based on the markers) and 2.56° ÷ 91.75° (0.04 rad ÷ 1.60 rad, based on the IMUs) for the dampers prototype and approximately 7.33° ÷ 109.89° (0.13 rad ÷ 1.92 rad, based on the markers) and 7.41° ÷ 107.06° (0.13 rad ÷ 1.87 rad, based on the IMUs) for the cam prototype. It can therefore be concluded that the knee angle results from both methods are very similar.

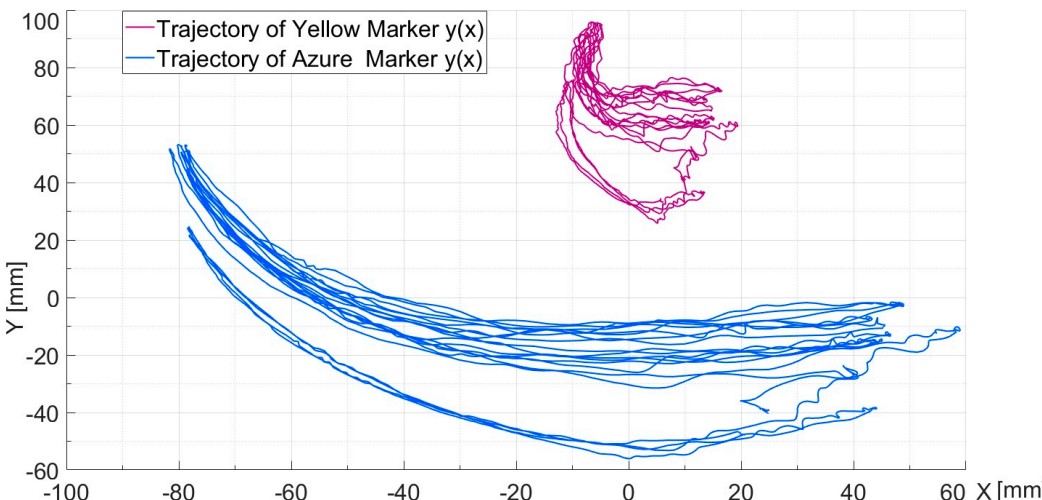

**Figure 17.** Y(x) as movement trajectory of azure and yellow markers—cam prototype.

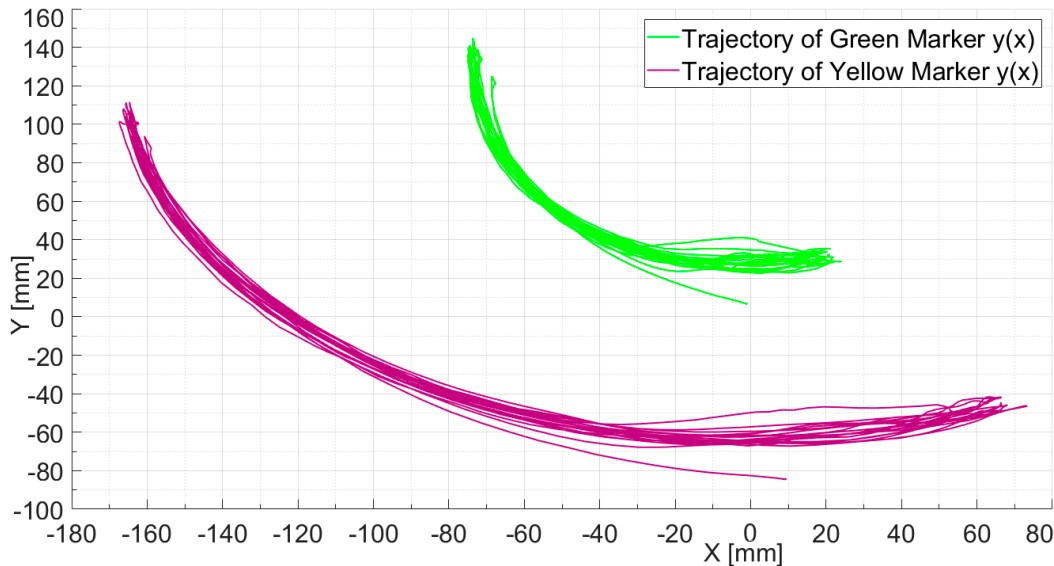

**Figure 18.** Y(x) as movement trajectory of green and yellow markers—dampers prototype.

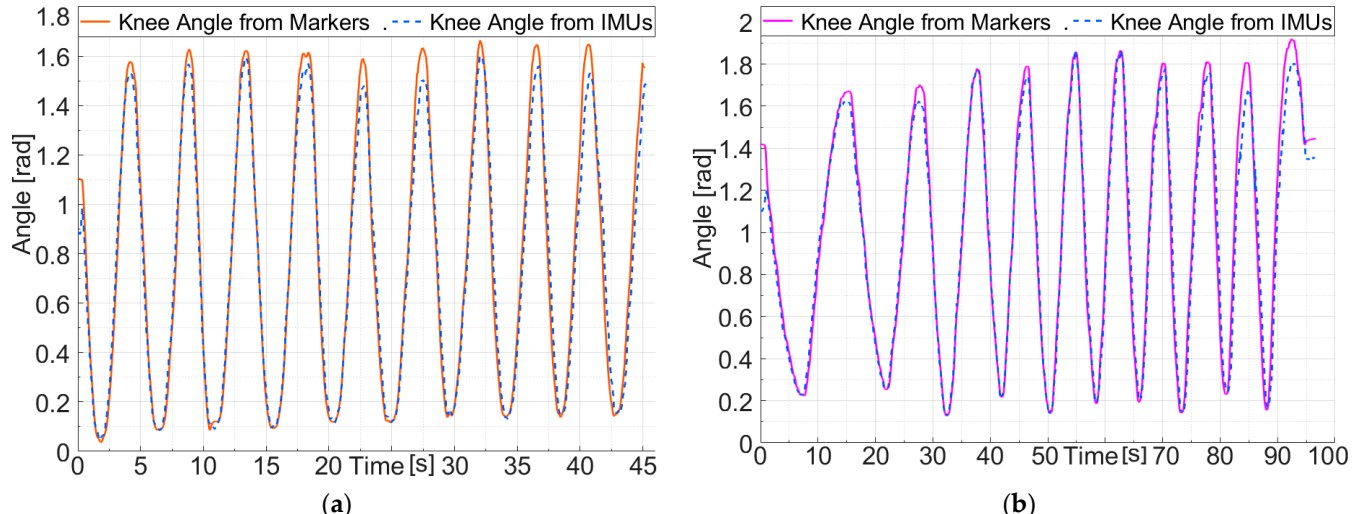

**Figure 19.** The flexion/extension angle of the knee joint in the sagittal plane determined by measuring the movement of the colour markers and IMU sensors: (**a**) cam prototype; (**b**) dampers prototype.

As a summary, Table 3 presents the numerical results of the obtained (from the colour markers) knee joint angle in the sagittal plane, including the min. and max. values, the ROM and the variations observed between the repetitions of the flexion/extension movement. As can be seen, the variations of the ROM values do not exceed ±0.078 rad for each prototype.

**Table 3.** The numerical data of the knee joint angle (from the markers) for the experiments with volunteers.

| No. | Prototype | Full Extension Angle [rad] | | | Full Flexion Angle [rad] | | | ROM [rad] | | |
|---|---|---|---|---|---|---|---|---|---|---|
| | | Min | Max | Variation | Min | Max | Variation | Min | Max | Variation |
| 1. | Dampers prototype | 0.045 | 0.142 | ±0.049 | 1.578 | 1.664 | ±0.043 | 1.471 | 1.542 | ±0.036 |
| 2. | Cam prototype | 0.129 | 0.244 | ±0.058 | 1.700 | 1.863 | ±0.082 | 1.562 | 1.718 | ±0.078 |

## 4. Discussion

The main objectives of the paper are two-fold: to demonstrate two designed mechanisms that reproduce the complex movement of the human knee and to present the results of tests carried out using the constructed prototypes. For these mechanisms, the author has defined the kinematics, geometry and shapes of the cams and has developed their three-dimensional models using CAD software. The prototypes were assembled mainly using parts manufactured with additive technology.

The first is the dampers prototype based on a modified crossed four-bar mechanism, that is a linearly adjustable four-bar mechanism, which is obtained by introducing two additional degrees of freedom in the form of prismatic joints for the crossed bars. It should be mentioned that the shock absorbers used in the prototype allow it to adapt to different conditions and achieve a variety of desired trajectories, constituting one of its main advantages. However, at this stage of the research the device lacks the ability to actively control the lengths of its dampers. This limitation restricts the adaptability and potential of the adjustable four-bar mechanism, but was accepted for the initial prototype due to design simplicity and to allow for a preliminary evaluation. Therefore, future plans need to include integrating actuators to dynamically adjust the dampers' lengths.

The second is the cam prototype with two cooperating cams (a 2DOF kinematic pair) as the main components, which makes it possible to achieve a precise and complex trajectory. Moreover, the application of cam profiles based on the ICR trajectory rather than on bone shapes means that movement can be reproduced without slippage. The ties ensure that the cams work closely together, rotating in succession, and limit the mechanism to 1DOF. Another key feature is that the mechanism can easily and repeatedly be modified to meet the requirements of different individuals by achieving different trajectories. The cams are interchangeable 3D-printed parts that are attached to the tibia and femur, respectively. They can be exchanged for parts with different profile shapes to suit each individual person.

Table 4 presents a comparison of the proposed prototypes with a few existing devices. Since the primary focus of this research is on knee joint prototypes, only a limited number of examples belonging to multi-joint devices, e.g., [36,37], is included. Both proposed prototypes demonstrate innovative approaches to knee joint mechanism design and offer distinct advantages over currently existing devices in that they can reproduce the complex movement of the human knee in the sagittal plane, despite being reduced to a two-dimensional movement. Many solutions, often simplify knee motion to a hinge, neglecting the complex nature of knee movement. These are found in exoskeletons like the knee–ankle–foot robot [38], Lokomat [39], Lopes [40], Hal [41], BLEEX and its successors, Ekso [42] and many other supporting and rehabilitation devices provided in [9,36,37]. While some devices, such as the Kin-Com [25], attempt to closely mimic knee joint movement, they often rely on fixed geometries, like four-bar linkages [24,26,28,30]. However, these might only approximately realize the exact movement and their adjustability is limited, potentially restricting their applicability to a narrower range of patient anatomies. The proposed dampers prototype, with its adjustable four-bar mechanism, offers a potential solution to these limitations. By incorporating real-time control and variable dimensions, it allows for greater adaptability and personalization compared to four-bar mechanisms with a rigid geometry. The cam prototype, on the other hand, exploits the concept of individual ICR trajectories. By directly incorporating ICR data into the design of the cam profiles, it offers a unique approach to achieving personalized knee movement. This strategy provides a more direct and potentially more accurate representation of individual knee kinematics than those often inspired by bionic principles [28,43,44] and relying on bone shapes [32,45]. These devices, while offering high precision, require extensive customization, increasing complexity and potentially hindering widespread adoption. In

conclusion, the dampers prototype emphasizes adaptability and real-time control, while the cam prototype focuses on easily customizable personalized movement based on individual ICR trajectories. These features, combined with a focus on knee kinematics, have the potential to improve a patient's comfort, reduce pain and enhance the effectiveness of rehabilitation and assistive devices.

**Table 4.** A comparison of the developed prototypes with several existing devices.

| Name | Joints Supported/Knee Joint DOF | Knee Joint Mechanism | Knee Joint ROM | Adaptability to Individual User | Application |
|---|---|---|---|---|---|
| Chen2016 [38] | 2/1 | Hinge joint | 60° | NA * | Gait training |
| Lokomat [39] | 3/1 | Hinge joint | NA | Adaptable to subjects with a range of femur lengths and pelvic widths | Gait rehabilitation |
| Kin-Com [24] | 1/1 | Based on the Chebyshev's linkage structure | 135° | Limited—dimensions optimized for a specific case | Rehabilitation |
| RoboKnee [46] | 1/1 | Hinge joint | NA | Limited—requires adjustments to control parameters | Performance augmentation |
| Gao 2021 [30] | 1/1 | Crossed 4-bar linkage | 120° | Limited—4-bar mechanism optimized for specific knee movement | Rehabilitation |
| DAEQOUS-G1 [32] | 1/2 | Articulating surfaces based on natural knee | 90° | Strictly individual design for each person | Endoprosthesis |
| Lovasz 2014 [20] | 1/1 | Geared linkage | 120° | Limited—designed with dimensions for a specific case | Prosthesis |
| Dampers prototype | 1/3 | Linearly adjustable crossed 4-bar mechanism | 120° | Real-time adjustment of ICR trajectory to individual needs | Rehabilitation |
| Cam prototype | 1/1 | Cam joint with ties and a spring | 120° | Interchangeable cams with individually customizable profiles | Orthosis, prosthesis |

* NA: the information is not available.

The originality of the new solutions could be further accentuated by applying different characterisation methods and comparing the solutions with even more existing devices, e.g., those presented in [47]. However, some features like the accuracy of the device or user comfort are difficult to obtain/asses. Nonetheless, the prototypes, with the ability to closely mimic physiological knee kinematics, have the potential to enhance patient comfort and reduce pain levels. The utilization of cam profiles derived from individual ICR trajectories enables personalized knee movement patterns. Moreover, the dampers prototype potentially offers a real time and very precise adjustment of the ICR trajectory adapted to individual needs, thereby improving the rehabilitation process, because the correct trajectory can be corrected step by step. Therefore, this kind of system can be very useful in the treatment of soft injuries (twists or ligament damage) when the biomechanics and mobility of the knee joint are disrupted and need careful direction. The initial real-life applications could include rehabilitation, as well as supporting gait and stand-sit sit-stand movements. Moreover, when an actuator-powered device is utilized with people, one of its primary operating modes would be the ability to move in accordance with a predetermined function of flexion/extension. However, future plans include adding sensors in order to create a mechatronic device that can function in different modes. These might include a collaboration mode, which involves identifying intentions, following the patient's

movements and providing the appropriate level of assistance or resistance, which involves developing strength by maintaining a predetermined load level. Overall, the mechanics of the prototypes can be used in exoskeletons, orthoses, prostheses, movement modelling and walking robots for various applications, including supporting agriculture.

The proposed prototypes were tested in several cases, assessing the correctness of the reproduced movement and the capabilities of the mechanisms, although research with human volunteers was the main focus of this article. As experimental characteristics of the prototypes, the femur and tibia trajectories were analysed in detail and the results are presented in graphs. To determine the movements of the mechanisms, video recordings were analysed to track the colour markers that were placed on the prototypes. In addition, IMU sensors were used as a supplement.

Firstly, for the tests carried out on a stationary rig, the consecutive experimental movements, for the marker displacements, trajectories and joint's angular ROM showed a reasonable level of comparability (for min/max positions the variation did not exceed ±0.3 mm). It was assumed that satisfactory accuracy is an about 0.5 mm variation in the minimum and maximum locations of the markers between successive movement repetitions. In this case, the extreme positions obtained by the mechanism (corresponding to the minimum and maximum flexion/extension) and the repeatability of the movement were considered to be the main determinants of satisfactory results. To allow for more accurate measurement, the movement of the prototype was stopped for 0.5 s in these extreme positions. These factors, together with the fact that the experimental movement was repeated ten times, made the results satisfactory and the characteristics of the devices' performance could then be determined.

The proper repeatability of movement, although lower than with the stationary rig, was also obtained in the trials with volunteers. This applies to flexion/extension movement and the knee's angular ROM (Table 3—variation of ROM ±0.036 rad for the dampers prototype and ±0.078 rad for the cam prototype), as well as the displacement and trajectory of the tibia's markers. Moreover, very similar results (with a max. 3° difference) for the knee angle were obtained from both measurement methods (markers and IMUs), showing a comparable error level and the possibility of reaching the future aim—replacing markers with IMUs to obtain a portable independent mechatronic device.

In general, lighting conditions were found to have a significant impact on the recording analysis results. Certain colours and shades should be avoided or replaced by other types of markers (such as QR codes), as these were not always correctly recognised. This can lead to increased measurement accuracy and a reduced time of postprocessing. However, the colour identification tests were also performed to assess the method's potentially wider applications. The recording frame rate (30 frames/s) and the resolution (1920 × 1080 pixels) were other factors that limited the precision of the measurement. During the trials, the recorded variations of a stationary marker position were less than ±0.1 mm apart. This might be taken as the precision of the video measurement and the postprocessing technique used, but needs further investigation. Moreover, for instance, during trials of the cam prototype at the stationary rig, the green marker's average speed was around 0.71 mm/frame, which indicates that there may be some inaccuracy in the measurement during movement at the given recording speed of 30 frames/s.

Irrespective of its limitations, the video analysis method proved to be repeatable and reliable. However, since the trials were carried out with two volunteers, a more thorough statistical analysis of the results and errors (including, e.g., standard deviation) should be performed in future experiments with a larger group of volunteers to improve statistical validity, data consistency and confidence. To evaluate the generalizability and obtain more representative results, it is necessary to check the prototypes for a wider range of users, including variations in height, weight and body proportions. This would make it

possible to assess the prototypes' sensitivity to these variations and evaluate them across a more diverse population. Due to the limited number of trials, further prolonged studies with many participants could also help in assessing the prototypes durability, which would be crucial for real-life applications. To confirm the extent of support offered by the prototype and establish any potential additional burden on the user, studies could also include powered prototypes with volunteers and/or EMG sensors, like in [48]. This would enable comparison of muscle use during knee movement with and without the prototype and offer a more comprehensive evaluation.

For the reasons mentioned above, including the limitations of the applied video analysis method, further research is needed to determine the system's suitability for the human knee joint. A comparison of the results with those from an advanced measurement system, such as a commercial optical motion capture system that uses multiple cameras (like Vicon [9] or Contemplas [4]), could enable a more precise evaluation of the results and could confirm the accuracy of the system. This should be incorporated into future studies with a larger group of volunteers to provide more accurate data for evaluation. Future plans also include tracking the ICR trajectory of the devices and the knee joint using IMU sensor data and video marker analysis.

Achieving fully ready mechatronic devices would still require some design improvements, the development of a control strategy and performing clinical trials. For each prototype, these could include future steps such as:

- Enhancing stiffness and durability by using robust materials for the device's frame;
- Applying a mounting system that ensures stability, safety and comfort for the user (e.g., with the application of soft materials);
- Implementing drives to actively control and adjust dampers' lengths in real time (dampers prototype-specific);
- Using more stretch-resistant material for the ties and applying springs with higher stiffness to better ensure constant contact between cams (cam prototype-specific).
- Integrating motors and sensors (encoders, IMUs and force sensors) to enable the precise monitoring of mechanism operation;
- Applying a control strategy enabling intuitive interaction with the device (real-time adaptability) in response to the user's movements (e.g., following the motion);
- Conducting clinical trials with a larger group of users to verify the device's effectiveness, safety and to identify potential problems in various real-life applications and settings.

The proposed suggestions provide a solid foundation for the further development of the knee joint prototypes. Their implementation will enable the creation of more advanced and functional devices that could improve the quality of life for many people.

## 5. Patents

Kiwała, S.J.; Gronowicz, A.; Ceccarelli, M.; Olinski, M. Mechanism for a knee prosthesis (Polish). Poland P 426870, 04.09.2018.

Olinski, M.; Handke, A.; Kiwała, S.J.; Wudarczyk, S. Mechatronic trajectory modification system, especially for knee prosthesis (Polish). Poland P 419889, 20.12.2016.

**Funding:** The preparation of this publication was supported by the pro-quality subsidy for the development of the research potential of the Faculty of Mechanical Engineering, Wrocław University of Science and Technology, in 2024 (number 8211204601).

**Informed Consent Statement:** Informed consent for publication was obtained from all identifiable human participants.

**Data Availability Statement:** Restrictions apply to the datasets due to privacy reasons. The datasets are not readily available because the data are part of an ongoing study and are subject to time limitations. Requests to access the datasets should be directed to the corresponding author.

**Acknowledgments:** The author thanks the staff/students at LARM, Cassino, Italy.

**Conflicts of Interest:** The author declares no conflicts of interest. The funders had no role in the design of the study; in the collection, analyses or interpretation of the data; in the writing of the manuscript; or in the decision to publish the results.

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
