# Peer review of "Design and Experimental Characterization of Developed Human Knee Joint Exoskeleton Prototypesâ€"

_machines, doi:10.3390/machines13010070_

Round 1
Reviewer 1 Report
Comments and Suggestions for Authors
The paper presents an interesting study on mechanisms equivalent to the knee joint in certain aspects. The mechanisms were designed and tested in the flexion-extension motion. The results have practical implications and the study is in line with the scope of the journal. That being said, I have some minor suggestions to consider for the Author for improving the paper further.
Introduction
1. It seems that the paper is mostly focused on the tibiofemoral part of the knee. Maybe this should be emphasized more, as the knee also contains other subjoints, such as the patellofemoral part.
2. The aim of the study and its novel aspects should be more clearly stated at the end of the Introduction. Lines 72-74 should be revised (this part is done better in Discussion - lines 274-276).
Results
3. Line 208: wearying -> wearing
4. Consider redoing Fig. 14 and 15 so that the angle is in deg and 0 represents full extension.
Author Response
"Please see the attachment."

Reviewer 2 Report
Comments and Suggestions for Authors
This manuscript presents two knee joint mechanism prototypes to replicate the human knee's complex movement. The prototype uses a linearly adjustable crossed 4-bar mechanism, while the second uses a cam mechanism. Both prototypes were built using 3D-printed parts and experimentally evaluated using video analysis and IMU sensors to track flexion/extension movements. Testing was conducted in both stationary frames and on volunteers. However, after reading carefully, following concerns are noted.
1. The author should revise the title for 'knee joint prototype' to 'knee joint exoskeleton' because the knee joint prototype indicates a biomechanical knee as a replacement for an actual human, which is not the case here.
2. The first prototype’s adjustable 4-bar mechanism lacks any control over damper lengths, limiting its adaptability. Adding actuators or control mechanisms to adjust damper lengths could enhance the prototype's versatility.
3. The study's human testing is limited to only two volunteers, which restricts the generalizability of the findings. A larger sample size would provide more statistically significant data and improve confidence in the results.
4. The video analysis suffers from limitations due to lighting conditions and color misidentification of markers, impacting data accuracy. The authors should discuss the limitations and add utilizing more advanced tracking systems (e.g., commercial motion capture systems) in the future.
5. The current study lacks consideration for real-time applications, which is crucial for developing prosthetics or assistive devices. Evaluating the prototypes under real-time conditions could reveal practical limitations not observed in the current setup. The authors should include more relevant references to single and multi-joint rehabilitation devices for lower limbs. See this work "Development of active lower limb robotic-based orthosis and exoskeleton devices: a systematic review."
6. The prototypes have not been tested for robustness or durability, especially under repeated or prolonged use. Future studies should include durability testing to determine how well these prototypes hold up under continuous use.
7. The manuscript does not consider the impact of using these prototypes on muscle activation and engagement, which is vital for assessing the devices' potential benefits or drawbacks. Please add including EMG analysis in future studies would offer a more comprehensive evaluation.
8. The manuscript lacks a detailed comparison of these prototypes with existing knee joint mechanisms, making it difficult to assess the novelty and potential advantages. The authors should include a comparative analysis with other designs that would strengthen the manuscript’s contribution to the field.
Author Response
"Please see the attachment."

Reviewer 3 Report
Comments and Suggestions for Authors
This paper presents a novel design and experimental characterization of two knee joint prototypes intended to simulate the complex movement of the human knee.
Comments are:
1.The literature review is somewhat limited, with insufficient discussion on how these prototypes differ from or improve upon existing models of knee joint simulation. Incorporating more comparative analysis with other biomechanical knee models or exoskeleton prototypes would strengthen the context and underscore the innovation.
2.The descriptions of the two prototypes lack detailed specifications, such as dimensions, materials, or critical tolerances. Providing a technical table summarizing these parameters would greatly aid comprehension.
3.The paper does not address measurement uncertainties or the calibration details of the IMU sensors and video analysis system, which are critical in motion analysis. Including a section on measurement validation or error analysis would enhance the credibility of the results.
4.The sample size of two volunteers is too small to generalize the results. Future studies should aim for a larger sample to improve statistical validity. Additionally, information on volunteer demographics (e.g., age, BMI) could help assess how well the prototypes might fit diverse users.
5.The results section is somewhat fragmented, with a heavy focus on visual data without sufficient accompanying quantitative analysis. For instance, it would be helpful to see the range of variation for each joint movement parameter across repetitions.
6.Some figures are difficult to interpret due to unclear legends or axes labels. Figures 12 and 15, in particular, could benefit from clearer annotation of data points and additional statistical markers (e.g., standard deviations) to indicate data consistency.
7.The discussion could better contextualize the findings with reference to the specific applications of these prototypes. How do these prototypes perform in comparison to existing devices in terms of range of motion, accuracy, or user comfort?
8.The limitations of the prototypes are mentioned briefly, but the future directions could be expanded to outline concrete steps for achieving full usability, such as enhancing durability or adding real-time adaptability in response to user motion.
Comments on the Quality of English Language
Some sentences are lengthy and may benefit from revision for clarity and conciseness. For example, technical descriptions should avoid ambiguous terms such as "reasonable accuracy" and instead provide quantitative benchmarks wherever possible.
Additionally, minor grammatical errors are present, and the paper would benefit from a final proofreading pass to correct typographical errors.
Author Response
"Please see the attachment."

Round 2
Reviewer 2 Report
Comments and Suggestions for Authors
The authors have addressed all the concerns raised by the reviewer. The manuscript can now be accepted.
Reviewer 3 Report
Comments and Suggestions for Authors
Proofreading.
Comments on the Quality of English LanguageProofreading.